# Human Exposure Assessment to Wearable Antennas: Effect of Position and Interindividual Anatomical Variability

**DOI:** 10.3390/ijerph19105877

**Published:** 2022-05-12

**Authors:** Silvia Gallucci, Marta Bonato, Emma Chiaramello, Serena Fiocchi, Gabriella Tognola, Marta Parazzini

**Affiliations:** 1Institute of Electronics, Computer and Telecommunication Engineering (IEIIT), National Research Council (CNR), 20133 Milano, Italy; marta.bonato@ieiit.cnr.it (M.B.); emma.chiaramello@ieiit.cnr.it (E.C.); serena.fiocchi@ieiit.cnr.it (S.F.); gabriella.tognola@ieiit.cnr.it (G.T.); marta.parazzini@ieiit.cnr.it (M.P.); 2Department of Electronics, Information and Bioengineering (DEIB), Politecnico di Milano, 20133 Milano, Italy

**Keywords:** wearable device, EM fields, human exposure

## Abstract

(1) Background: This work aims to assess the human exposure to the RF-EMFs emitted by a wearable antenna. (2) Methods: a wearable antenna tuned at f = 2.45 GHz was tested by placing it in six realistic configurations relative to a male and female human model. The exposure assessment was performed by means of computational methods to estimate the SAR_10g_ distributions at 1W of input power. (3) Results: (i) for all the configurations the SAR_10g_ distributions resulted always mainly concentrated on a superficial area immediately below the antenna itself; (ii) the obtained values have shown that the configuration with the highest exposure value was when the antenna was posed on the arm; (iii) the exposure tends to be higher for male model. (4) Discussion and Conclusions: This work highlights the importance of performing an exposure assessment when the antenna is placed on the human wearer considering the growth of the wearable technology and its wide variety of fields of application, e.g., medical and military.

## 1. Introduction

The interest in wireless applications in the communication environment is increasing in recent years with particular focus on wearable and textile antennas. One of the most important characteristics of wearable antenna is the wide variety of applications they lend themselves to: mobile communications, wireless medical, and military applications, and, for this reason, there is not a standard for this type of antenna [1,2,3,4]. Furthermore, the frequency spectrum involved in this technology is huge: it ranges from the frequencies typical of the GSM communications up to the 5G spectrum [5]; the diffusion of this new technology is widespread: smart watch, medical sensors, smart clothing, smart glasses, wearable camera and so on [6].

Since the wearable devices are often integrated into the user’s clothing, the wearable antenna must be lightweight, thin, easily integrated into the garment, and it should not obstruct the user’s movements [7]. Moreover, the position of the antenna relative to the human body is a crucial aspect since, because of that, the wearable device may vary from its original setting and consequently, its performances could change. For example, the bending effect of the antenna can often occur, and it has been deeply studied [2,8] revealing that it could detune the antenna, compromising its correct functions [9].

Due to the very close position of the antenna to the user’s body, the assessment of the human exposure to the electromagnetic field (EMF) emitted by wearable devices is needed. Until now, most of the studies have been focused on the assessment of the EMF human exposure in specific configurations of the antenna with respect to the human body; for instance, Gharode et al. [5] have reviewed the literature regarding the assessment of the EMF emitted by the wearable devices showing that most of the reported articles evaluated the EMF exposure mainly considering a single human model or a simplified phantom, made of three or four layers (e.g., [10,11]) reproducing the layers of a generic superficial part of the human body. Furthermore, the analysis of the interactions between the antenna and the human body is often addressed with the aim to study if and how the performances of the antenna vary in the presence of the human body [12].

For all these reasons, the aim of this work is specifically to assess the exposure of the human being to the EMF emitted by a wearable antenna belonging to the typical spectrum of the wearable devices: it is tuned at f = 2.45 GHz, that is the characteristic frequency of the ISM (Industrial, Scientific, and Medical) band, used for the devices designed for the health monitoring of the user. A typical field of application of this type of wearable devices is the military environment [13] since the physiological parameters of the soldier during a mission, monitored and scanned by the wearable device, are important data to be communicated. Therefore, we will refer to this occupational exposure situation in the following of the study.

The proposed study takes place in this context and our aim is to further broaden the knowledge on the assessment of the exposure of the human being to the EMF emitted by a wearable antenna. More in details, EMF human exposure assessment has been evaluated by means of electromagnetic computational method considering the wearable antenna posed in six different positions on the human body, mimicking different realistic exposure conditions. This will allow to evaluate the effect of the antenna position on the human exposure, in order to understand if and how the variation of the involved anatomical district could significantly influence the exposure. Moreover, to evaluate how the interindividual anatomical variability could impact on the level of exposure, two different human models were used, one of an adult male and one of an adult female. Indeed, there are differences in the anatomy models depending on the gender that could impact on the exposure level. For all the simulated conditions, the human exposure levels were assessed and compared by evaluating the specific absorption rate (SAR), the quantity indicated in the ICNIRP guidelines [14]. More in specific, the distributions of SAR averaged over 10-g cubic mass were evaluated in some specific tissues.

## 2. Materials and Methods

In order to assess the human exposure to the EMF emitted by wearable antennas, such an antenna for on-body communication has been simulated, tuned at f = 2.45 GHz. The first step of the study consisted in simulations with the antenna in free space, then the simulations have been performed by placing the antennas near a human model in different positions. The antenna has been simulated with both human models, male and female adult model of the Virtual Family.

### 2.1. Antenna Structure

The simulated antenna (Figure 1) is tuned at f = 2.45 GHz, and it is based on the work of Chahat et al. [15]. This source is a coplanar-fed antenna, printed on a polyethylene foam (σ = 0.0005 S/m, ε_r_ = 2.25) of 3 mm of thickness. Furthermore, the radiating elements and the ground plane are made of copper (σ = 5.81 × 10^7^ S/m). The overall size of the antenna is 25 × 38.5 mm. Table 1 summarized the detailed dimensions of the antenna; it is noteworthy that the thickness of the patch is infinitesimal. Figure 1 shows the radiation pattern and the reflection coefficient of this antenna, showing a good agreement with the results of [15].

### 2.2. Exposure Simulations

All simulations were implemented with the finite-difference time-domain (FDTD) solver of the Maxwell’s equations through the approximation to finite differences. Briefly, the FDTD method involves both a spatial and temporal discretization of the electric and magnetic fields over a period of time and a specific spatial domain limited with the boundary conditions. Typically, the minimum spatially sampling is at intervals of 10–20 per wavelength, and temporal sampling is sufficiently small to maintain stability of the algorithm [16,17]. All the simulations were performed in the platform Sim4life (ZMT Zurich Med Tech AG, Zurich, Switzerland, www.zurichmedtech.com, accessed on 28 April 2022). The human models Duke, an adult male (age = 34 years old, height = 1.77 m, mass = 70.3 kg, BMI = 22.4 kg/m^2^) and Ella, an adult female (age = 26 years old, height = 1.63 m, mass = 57.3 kg, BMI = 21.5 kg/m^2^), from the Virtual Population [18] were used.

The antenna was placed in several positions relative to the human model, according to both the most common and possible uses of these devices: (i) on the ankle, (ii) on the arm, (iii) on the leg, (iv) on the shoulder, (v) on the torso at the height of the heart, and, finally, (vi) on the head. In each simulation, the wearable antenna with the side of the substrate turned towards the human model was positioned at a distance from the human model of 2 mm except for the head’s case in which the distance is almost equal to the thickness of the helmet (10 mm). Indeed, when the antenna was positioned close to the head, the antenna has been placed outside a helmet, worn by the user. The helmet was simulated made of steel (σ = 1.1 × 10^6^ S/m, ε_r_ = 1). Only in the exposure scenario with the antenna near to the head, the wearable antenna has been bent in order to better adapt the shape of the antenna to the curvature of the helmet. All of these configurations have been implemented on both human models (Ella and Duke).

The boundary conditions were set as absorbing conditions with the Perfect Matched Layer (PML) and the dielectric properties of each tissue of the human models were selected according to the literature [19,20]. Furthermore, the mesh was set with step ranging from 0.2 mm up to 2 mm in order to discretize all model tissues correctly according to the working frequency.

Figure 2 shows, as an example, the positions of the antenna here analyzed on Duke. Furthermore, the figure shows how for each antenna position the computational domain was limited to a specific region of interest of the human body, which was supposed to be the most exposed.

### 2.3. Exposure Assessment

In order to evaluate the interactions between the EMF emitted by the simulated wearable antenna and the human tissues, the SAR was calculated; it can be defined as the electromagnetic energy absorbed by a human tissue. With more details, the SAR is defined as the time derivative of the incremental energy consumption by heat involved in an incremental mass in a volume element, characterized with its density ρ and its conductivity σ [14]. More specifically, the distribution of SAR value averaged on a cubical mass of 10 g (indicated from now on as SAR_10g_) was estimated with the input power of the antenna set at 1 W. The values of the SAR_10g_ distribution have been extracted considering a cubical box with sizes of 250 × 250 × 250 mm in all the simulations except the scenario in which the antenna was placed on the head of the human model; in that case the data has been extracted from the entire domain, however limited to the head area. The box was always centered on the antenna. The size of the cubical box was chosen to be sufficient to include all significant SAR_10g_ values, including the first slices of the human tissues in which the calculated SAR_10g_ was null. Furthermore, to better quantify the SAR_10g_ distributions, the descriptive statistics (minimum, 25th, 50th, 75th percentile, and maximum) of the SAR_10g_ distributions in the tissues included in the cubical box were calculated. Finally, the peak SAR_10g_ have been compared with the ICNIRP Guidelines [14] limits for the occupational exposure in the frequency range of 100 kHz to 6 GHz.

## 3. Results

In this section, the results regarding the assessment of the EMF exposure due to the wearable antenna are collected.

In Figure 3, the distributions of the SAR_10g_ in two of the simulated antenna positioning are shown as an example. Specifically, panels in the upper part of the figure shown the SAR_10g_ distribution over the skin when the wearable antenna tuned at f = 2.45 GHz (Figure 3a) is posed near to the Ella ‘s shoulder, whereas panel in the lower part shown the SAR_10g_ distribution over the skin when the wearable antenna tuned at f = 2.45 GHz (Figure 3b) is posed on the ankle’s Duke. All the values are normalized with respect to the maximum SAR_10g_ value found in each configuration. A qualitative evaluation of the panels in Figure 3 indicated that the regions of higher SAR_10g_ were mainly concentrated in the portion of the body closer to the antenna, just below the antenna itself, with values of the SAR_10g_ distributions that tends to decrease in the farther regions of the body, as expected.

To better characterize the SAR_10g_ distributions in the region of interest, Figure 4 shows the descriptive statistics (minimum, 25th, 50th, 75th percentile and maximum) of the SAR_10g_ distribution evaluated in the tissues included in the cubical box identified in the anatomical district of interest with respect to the antenna positioning. From the top to the bottom of Figure 4, the panels are relative to Duke (Figure 4a) and Ella model (Figure 4b). All the panels clearly show that, for all the configurations, there is a great gap between the maximum value and the 75 percentiles of the distribution. As a general trend, it is therefore reasonable to assume that the large part of the values of the SAR_10g_ distributions obtained are well below their peak values.

This aspect was further investigated calculating the percentage of the values of the distribution of SAR_10g_ greater than or equal to the 90% of the peak value of the distribution itself, for each configuration. Table 2 collects these percentage values. Data clearly shows that, on average, only 2% of the SAR_10g_ distribution values exceeds the 90% of the peak value except when the wearable antenna was posed close to the ankle where this value can reach about 5% of the data, specifically for Duke model. This means that these distributions are highly narrowed around their peak values.

In order to analyze the effects on the EMF exposure due to the antenna position and interindividual anatomical variability, in Table 3 the peak values of the SAR_10g_ are collected for each configuration. From the table, it is noticeable that the minimum peak SAR_10g_ value was found when the antenna was placed close to the head on the top of a steel helmet, for both the human models. These values are clearly due to the presence of the helmet between the antenna and the human models. On the other hand, considering all the other configurations, the maximum value was found when the antenna was placed on the Duke’s arm. As to the effect of interindividual anatomical variability, the peak SAR_10g_ were higher in the case of Duke when the antenna was placed close to the arm, leg, and shoulder, whereas they resulted higher in the Ella model when the antenna was place close to the ankle.

Moreover, a statistical analysis showed that the differences between the distributions of the SAR_10g_ values of the male and the female models for each position of the wearable antenna resulted statistically significant (*p*-value < 0.01).

Interestingly, if we evaluate the SAR_10g_ distributions in some specific tissues included in the cubical box identified in the anatomical district of interest, the tissue with the highest peak SAR_10g_ was always the skin, followed by the subcutaneous adipose tissue and the muscle.

## 4. Discussion

This work aimed to assess the human exposure to the RF-EMFs emitted by a wearable antenna since, in recent year, we have been witnessing a growth of this technology due to its wide variety of fields of application, e.g., from medical to the military one. In particular, the antenna tuned f = 2.45 GHz has been placed near to two human models, one female and one male, in different positions, mimicking their possible realistic use.

Results of this study clearly show that for both human models and in all antennas positionings, the region with higher SAR_10g_ values are mainly located in a surface area immediately below the antenna itself and the SAR_10g_ distributions are narrowed around their peak value. When the wearable antennas were positioned on the top of the head the SAR_10g_ values obtained were extremely lower than all the other positions because this is the only scenario in which a medium (steel helmet) was interposed between the source and the user so the helmet shields the head of the human being, strongly reducing the interactions between the radiation and the human tissues and therefore also the contribution of the power that would otherwise have been absorbed by them. In order to deepen the incidence of the presence of a metallic medium that could shield the EMF reducing the impact on the human tissues, further simulations have been performed. More in details, in these simulations the steel helmet has been removed and then the SAR_10g_ values have been compared with the values of the previous case. In Figure 5 the boxplots relative to the configurations with Duke and Ella, both without the steel helmet are reported.

The results reported in Figure 5 clearly show an increase of the values of SAR_10g_ if compared with the data obtained in presence of the helmet. This trend demonstrates that the presence of a metallic medium influences the radiation pattern of the antenna and, consequently, also the human exposure. However, the values in absence of the helmet are lower than the values from all the other configurations and the reason is the distance between the antenna and the human model; indeed, in these simulations without the helmet, the antenna is more distant than the previous configurations.

Among all the other different positions of the wearable antennas here analyzed, the highest exposure values were found when the antenna was posed on the arm in the Duke model, while for Ella model, the highest peak SAR_10g_ has been obtained when the antenna was positioned on the ankle. Excluding the antenna position close to the head, for Duke model the lowest value of peak SAR_10g_ was obtained when the antenna was placed on the torso at the height of the heart, while for the Ella model the lowest value of peak SAR_10g_ has been obtained when the antenna was posed on the leg.

Furthermore, it can be noted that the values of the SAR_10g_ are quite similar between Ella and Duke models when the antenna was posed close to the ankle. Indeed, although the data have the same order of magnitude, for arm, leg and shoulder antenna positions, the exposure was much higher for Duke than for Ella (28% for the case of the shoulder, 110.9% for the case of leg and 75.9% for the arm’s case). A possible reason of the increase in the Duke’s values is the greater amount of muscle in the male model with respect to the female one indeed, in this tissue, higher values of SAR_10g_ have been detected unlike the female model. The opposite situation occurs when the antenna was posed on the torso, for which the exposure resulted higher for Ella than for Duke (i.e., an increase of 8.1%). To explain this evidence, SAR_10g_ distributions have been evaluated in each tissue considered in the cubical box and it was evident that the value found in the Ella’s breast gives strong contribution to the increase of the maximum value.

For any configuration analyzed, the peak SAR_10g_ was mainly located in the most external tissues. This behavior was expected with such a high frequency because of the relationship of inverse proportionality between the frequency and the penetration depth.

The peak values here obtained have been compared with the ICNIRP Guidelines limits [14]. Into the guidelines, there are two different occupational limit values depending on the anatomical district category considered: for the local Head/Torso the limit is 10 W/kg and for the local limb is 20 W/kg. These values of local SAR must be considered as averaged over 6 min and over a 10-g cubic mass. The limit values were exceeded in several exposure conditions here simulated: (i) Duke’s exposure when then antenna was posed on the arm, the leg, and the torso; (iii) for Ella’s exposure when the antenna was posed on the torso, and (iv) when the antenna was posed on the ankle. However, it is important to underline that the input power has been here set at 1 W, which is higher than the realistic values of the power supply of the wearable antennas, which is in the order of tens of mW [21,22]. Therefore, all the peak SAR_10g_ values should be re-scaled accordingly, resulting therefore always in compliance with the regulation.

Moreover, the obtained values of the peak SAR_10g_ are in line with the data reported in the literature. In fact, Ali et al. [11] estimated the peak SAR_10g_ to be equal to 8.16 W/kg when a simplified human arm, made of skin, fat, muscle, and bone, was exposed to the radiation emitted by a wearable antenna tuned at f = 2.4 GHz with an input power of 0.5 W. Moreover, Chahat et al. [4] have exposed a rectangular parallelepiped homogeneous phantom to a wearable antenna at f = 2.45 GHz and the peak SAR_1g_ obtained for an input power of 1 W was 48 W/kg. Evaluating the average of the peak SAR_1g_ across all the simulations here implemented (data not shown), except for the antenna positioning close to the head, resulted in a SAR_1g_ of about 34 W/kg. Finally, Chahat et al. [1] have simulated an antenna tuned at f = 2.45 GHz on two heterogenous adult phantoms (Duke and Ella models); the peak SAR_1g_ obtained when the antenna was positioned on the model torso ranged from 18.7 to 30.7 W/kg for an input power of 1 W. Similarly, evaluating the peak SAR_1g_ from our simulations when the antenna was posed on the torso for a frequency of 2.45 GHz, it ranges from 26.4 to 29.2 W/kg for 1 W of input power.

However, it is noteworthy that in the interpretation of the results of EMF numerical dosimetry an intrinsic level of uncertainty, that can influence the reliability of the estimated SAR values, should be always taken into consideration. These uncertainties are difficult to be quantitatively estimated, but are qualitatively due to many factors, such as the reduced knowledge of dielectric properties of the human tissues, numerical limits of modelling the source and anatomical body, discretization error, accuracy, and choice of the algorithm for SAR calculation [23,24,25,26]. In any case, although with unknown level of uncertainty, the numerical approach provides important and additional information on the distributions inside biological system and organs that cannot be obtained only by experimental dosimetry [27].

## 5. Conclusions

In conclusion, this work has aimed to assess the exposure of the human being to the EMF emitted by a wearable antenna, tuned at f = 2.45 GHz, investigating the effect for the antenna position and the human anatomical variability. For all the configurations here analyzed, the higher SAR_10g_ values resulted always mainly concentrated on a superficial area immediately below the antenna itself. Moreover, these distributions are narrowed around their peak values and tends to flatten toward lower values in the farther anatomical regions.

Among the twelve different analyzed configurations, the worst configuration in terms of the peak of the SAR_10g_ is when the antenna was placed on the male arm. As to the effect of anatomical variability, the exposure was higher in the case of Duke when the antenna was placed close to the arm, leg, and shoulder, whereas it was higher in the Ella model when the antenna was placed close to the torso and to the ankle.

## Figures and Tables

**Figure 1 ijerph-19-05877-f001:**
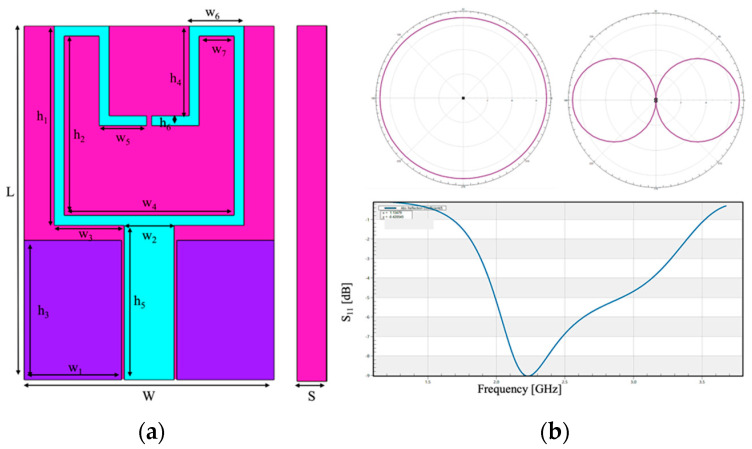
Simulated wearable patch antenna tuned at f = 2.45 GHz. (**a**) front view and the side view and (**b**) performance of the simulated antenna: on the upper part the radiation patterns of the antenna (on the left with phi = 0°, on the right with theta = 90°); on the lower part, the reflection coefficient centered on the frequency f = 2.45 GHz.

**Figure 2 ijerph-19-05877-f002:**
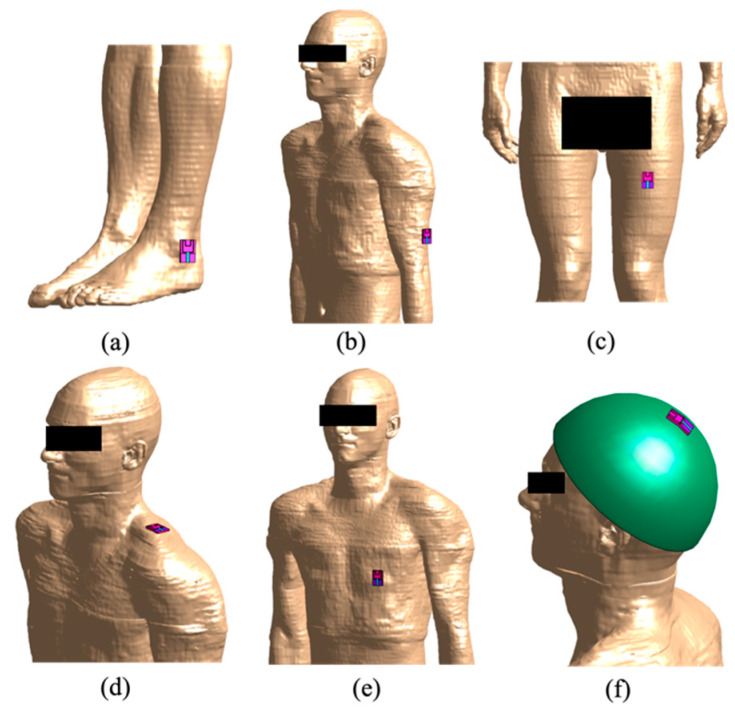
Overview of the six simulated exposure configurations on Duke for the antenna tuned at f = 2.45 GHz: (**a**) antenna on the ankle, (**b**) on the arm, (**c**) on the leg, (**d**) on the shoulder, (**e**) On the torso at the height of the heart and (**f**) on the head, placed on a steel helmet.

**Figure 3 ijerph-19-05877-f003:**
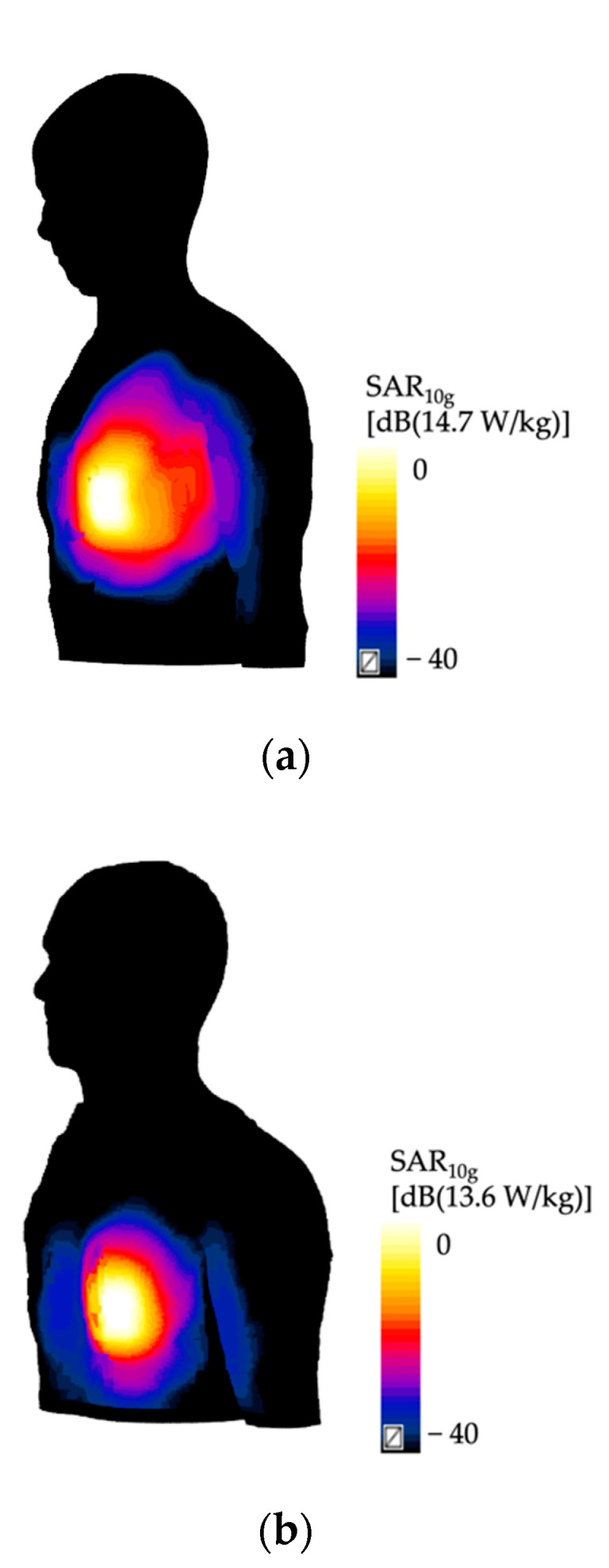
The distributions of the SAR_10g_ [W/kg/W] in two simulated configurations: (**a**) Ella with the antenna posed on the shoulder, and (**b**) Duke with the antenna placed on his trunk. The values are shown in decibel scale, with the reference value equal to the maximum value of the SAR_10g_ obtained in each configuration.

**Figure 4 ijerph-19-05877-f004:**
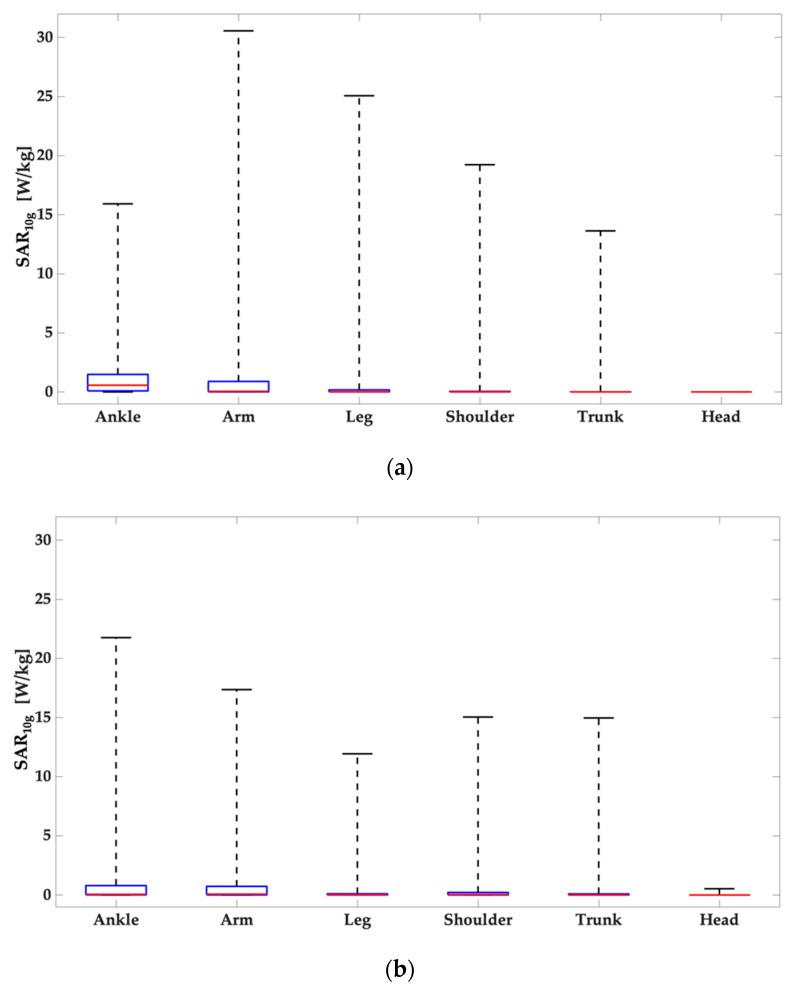
Descriptive statistics (minimum, 25th, 50th, 75th percentile and maximum) of the SAR_10g_ distributions in the tissues included in volume of interest relative to the analyzed anatomical district: (**a**) Duke and (**b**) Ella.

**Figure 5 ijerph-19-05877-f005:**
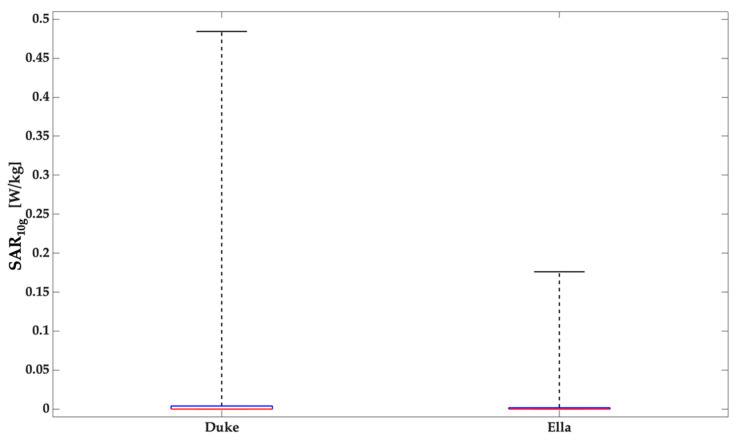
Descriptive statistics (minimum, 25th, 50th, 75th percentile and maximum) of the SAR_10g_ distributions in the tissues included in volume of interest relative to the analyzed anatomical district: Duke and Ella, both without the steel helmet.

**Table 1 ijerph-19-05877-t001:** Sizes of the antenna tuned at f = 2.45 GHz.

Values	Dimensions [mm]
L	38.5
W	25
S	3
h_1_	20
h_2_	18
h_3_	14
h_4_	9
h_5_	15.5
h_6_	1
w_1_	9.75
w_2_	5
w_3_	7
w_4_	17
w_5_	4.75
w_6_	5.5
w_7_	3.5

**Table 2 ijerph-19-05877-t002:** Percentage of SAR_10g_ values between 90% of the peak and the peak of the distribution.

Anatomical District	Duke	Ella
Ankle	4.7	2.2
Arm	4	2.6
Leg	2.5	1.1
Shoulder	1.5	2
Torso	1.1	1.1
Head	0.01	1.1

**Table 3 ijerph-19-05877-t003:** SAR_10g_ values extracted from the different exposure configurations [W/kg/W].

**Anatomical District**	**Duke** **f = 2.45 GHz**	**Ella** **f = 2.45 GHz**
Ankle	15.9	21.8
Arm	30.6	17.4
Leg	25.1	11.9
Shoulder	19.2	15
Torso	13.6	14.7
Head	0.0001	0.0001

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
