# Peer review of "Human Exposure Assessment to Wearable Antennas: Effect of Position and Interindividual Anatomical Variability"

_ijerph, 2022, doi:10.3390/ijerph19105877_

Round 1
Reviewer 1 Report
The Gallucci et al. article provides original and useful information in its documentation of the extreme reduction in SAR obtained by placing an antenna on a metal helmet, such as in Fig. 2 f and documented at the end of Table 3. It confirms quantitatively what we already know from extensive studies of cell phone SAR: that distance from the body is paramount in determining SAR.
What this means is that in military infantry, it will be likely possible to deploy helmet transmitters that will allow effective placement and communication with foot soldiers, and within a respectable range.
The 2 mm distance from the body used in modeling re-creates the same problem faced with cellular phones, where manufacturers can select the distance they wish in modeling. Can the antennas actually be nearer to the body than 2 mm, and what happens when the supporting tissue is drenched in sweat?
Unfortunately, the article does not follow the lead of cited works (10, 11) by Wissen et al. (2021) and Ali et al. (2017) who both used ground planes (high impedance metamaterial) to minimize SAR, which also stabilizes frequency tuning and improves gain by providing a surface wave band gap.
Although worn antennas are eyed initially for the military, they may in short term be transferred to civilian applications where SAR limits are 2 rather than 20 W/kg. There, we are facing longer term exposures that are simply not within the scope of the safety limits of ICNIRP, which are based on heat only, and on behavioral changes over 1 hour in monkeys and rats. Some health, rather than heat-based exposure limits have thresholds 3 orders of magnitude lower than heat-based limits. Also, the averaging over 10 g, as is done by ICNIRP, would not be credible outside the halls of engineering. The interaction of EMF with living systems is at the particle (electrons and protons) and molecular level, rather than 10 g. The publication of the engineering of body antennas that ignore elementary precautions to protect the wearer is not a service to technology.
Lines 113 and 114 are unclear.
Author Response
Point 1: The Gallucci et al. article provides original and useful information in its documentation of the extreme reduction in SAR obtained by placing an antenna on a metal helmet, such as in Fig. 2 f and documented at the end of Table 3. It confirms quantitatively what we already know from extensive studies of cell phone SAR: that distance from the body is paramount in determining SAR.
What this means is that in military infantry, it will be likely possible to deploy helmet transmitters that will allow effective placement and communication with foot soldiers, and within a respectable range.
The 2 mm distance from the body used in modeling re-creates the same problem faced with cellular phones, where manufacturers can select the distance they wish in modeling. Can the antennas actually be nearer to the body than 2 mm, and what happens when the supporting tissue is drenched in sweat?
Response 1: First of all, we would thank the reviewer for the useful revision of the manuscript and the valuable feedback provided. As to the minimal distance between the antenna and the human body, we have decided to set it equal to 2 mm because this value was considered appropriate to simulate the presence of the supporting tissue in which the antenna is integrated, and reasonable with respect to the range of distance between the wearable antenna and the human body used in literature, (i.e. Chahat et al., 2011 [15]; Smida et al., 2020 [21]). Moreover, we have kept fixed this distance in all the conditions and positions here analyzed to compare the exposure levels in all these different exposure scenarios. For sure, the distance between the source and the human body is an important parameter and decreasing the distance between the EMF source and the human model, it is reasonable to expect an increase of the values of the SAR10g. As to the supporting tissue drenched in sweat, this is a very interesting question that, unfortunately, here we have not considered, and it is out of the scope of our study. Indeed, here we were focused on investigating the effects on the exposure level of the interindividual anatomical variability and of the position of the antenna.
Point 2: Unfortunately, the article does not follow the lead of cited works (10, 11) by Wissen et al. (2021) and Ali et al. (2017) who both used ground planes (high impedance metamaterial) to minimize SAR, which also stabilizes frequency tuning and improves gain by providing a surface wave band gap.
Response 2: We thank the reviewer for the useful comment. In our study we have simulated a wearable antenna tuned at 2.45 GHz, following the one proposed in literature by Chahat and colleague (Chahat et al., 2011 [15]). In the “Discussion” session, we have compared our results with other data found in literature considering other wearable antennas working at the frequency of our interest, as, for example, the works cited by the reviewer. In some cases, these literature studies included in the antenna design also a ground plane aiming to reduce the exposure. This approach to compare our data with other found in literature was done to put our results in the context of the state of the art and to verify that our results were in line with other literature data.
Point 3: Although worn antennas are eyed initially for the military, they may in short term be transferred to civilian applications where SAR limits are 2 rather than 20 W/kg. There, we are facing longer term exposures that are simply not within the scope of the safety limits of ICNIRP, which are based on heat only, and on behavioral changes over 1 hour in monkeys and rats. Some health, rather than heat-based exposure limits have thresholds 3 orders of magnitude lower than heat-based limits. Also, the averaging over 10 g, as is done by ICNIRP, would not be credible outside the halls of engineering. The interaction of EMF with living systems is at the particle (electrons and protons) and molecular level, rather than 10 g. The publication of the engineering of body antennas that ignore elementary precautions to protect the wearer is not a service to technology.
Lines 113 and 114 are unclear.
Response 3: We thank the reviewer for the appropriate comment. In this study we have decided to focus our attention mainly on the exposure assessment due to a possible occupational use of this EMF source since at this moment their use is more common in the military than in the civilian sphere, but it is true that in the future their use also in the civil applications will grow. Anyway, our data can be also compared with the ICNIRP limit for the general public exposure. Moreover, it is important to underline that our data were obtained for an input power of 1 W (an easy way to normalize the results which is frequently used in this type of study), which is much higher than the realistic values of the power supply of the wearable antennas, which is in the order of tens of mW. Therefore, all the peak SAR10g values should be re-scaled accordingly, resulting therefore always in compliance with the regulation (both for occupational and general public exposure). As a final comment, in this study we have followed a typical and widely used approach to evaluate the exposure assessment which is the comparison with the regulation that are currently in use.
We have modified in the revised manuscript the sentence in lines 113 and 114 to make it clearer:
- Materials and Methods, 122: “the wearable antenna with the side of the substrate turned towards the human model was positioned at a distance from the human model of 2 mm except for the head’s case in which the distance is almost equal to the thickness of the helmet (10 mm)”.

Reviewer 2 Report
1: A small description of finite-difference should be added
2: Which one is the relevance in including female and male subjects in the study? it is not said.
3: "A possible reason for the increase in the Duke’s values is the greater amount of muscle in the male model with respect to the female one indeed, in this tissue, higher values of SAR10g have been detected unlike the female model". In my opinion, this asseveration needs a real experiment for its justification
4: This simulation is an excellent work, but it should be referenced in an experimental study.
5: An whole statistical analysis should be added. In particular, which one is the estimated error?
6: An additional text about the SAR meaning and its side effect on human tissue should be added
Author Response
Point 1: A small description of finite-difference should be added.
Response 1: Thank you for this comment. In the revised version of the manuscript, we have added the statement in which we explain that the FDTD method is used for the resolution of the Maxwell’s equations, also proposing the reference in which the method is well explained.
Materials and Methods, 103: “solver of the Maxwell’s equations through the approximation to finite differences. Briefly, the FDTD method involves both a spatial and temporal discretization of the electric and magnetic fields over a period of time and a specific spatial domain limited with the boundary conditions. Typically, the minimum spatial sampling is at intervals of 10–20 per wavelength, and temporal sampling is sufficiently small to maintain stability of the algorithm [16; 17].
Point 2: Which one is the relevance in including female and male subjects in the study? it is not said.
Response 2: First of all, we would thank the reviewer for the relevant comments. As to this specific comment, an aspect that is important to consider in this type of studies is the effect of the human variability on the exposure level. Therefore, we have decided to study both the case of the male and female models to evaluate this variability. Indeed, some anatomical differences, such as the breast in the female model or the mass of specific tissues (e.g., muscle) could have an impact on the assessment of the exposure due to wearable devices.
To better clarify this, we have added in the revised manuscript the following sentence:
- Introduction, 70: “there are differences in the anatomy of the models depending on the gender which could have an effect on the exposure level.”
Point 3: "A possible reason for the increase in the Duke’s values is the greater amount of muscle in the male model with respect to the female one indeed, in this tissue, higher values of SAR10g have been detected unlike the female model". In my opinion, this asseveration needs a real experiment for its justification.
Response 3: First of all, we want to thank the reviewer for the accurate revision and the valuable feedback provided, giving us the opportunity to deepen some relevant aspects. The cited sentence is a hypothesis that we have formulated to explain the evidence that the values of the SAR10g are higher in Duke’s model than in the Ella’s model. Indeed, taking a look on the structures and on the distributions of the tissues of both models, we have observed that the mass occupied by the muscle in the male model is greater than the female model (approximately 10 kg of difference as can be identified from the tissue distribution of the models). This evidence matches with our obtained results since the dielectric properties of the muscle are higher than those of the skin. More specifically, evaluating the dielectric properties of the muscle with respect to the skin, we have noted that both the conductivity (σ) and the relative permittivity (εr) of the muscle are higher than the ones of the skin so, it is reasonable to conclude the greatest presence in the volume of interest of the muscle in the male model than the female one, influences the SAR10g values, involving their increase. Anyway, we agree with the reviewer that this is just a hypothesis.
Point 4: This simulation is an excellent work, but it should be referenced in an experimental study.
Response 4: We thank the reviewer for the comment. In the original manuscript there are no references to experimental studies but with this comment we had the opportunity to justify our choice to perform computational studies. A branch of the dosimetry is the experimental one in which geometrically simplified phantoms filled with tissue-equivalent liquid are used and a real EMF source is placed close to them and then the measures of the electric field are performed when the source is on. This type of study is based on the approximation of the human body as a homogeneous phantom, omitting the differentiation of each single tissues and also the complex geometry of a human body. On the other hand, in the last 30 years, the computational dosimetry has become widely used in the exposure assessment studies and in the evaluation of the interaction between EMF and the biological tissues.
Indeed, numerical modelling of the interaction between electromagnetic fields (EMFs) and the dielectrically inhomogeneous human body provides a unique way of assessing the resulting spatial distributions of internal electric fields, currents, and rate of energy deposition. Knowledge of these parameters is of importance in understanding such interactions and is a prerequisite when assessing EMF exposure or when assessing or optimizing therapeutic or diagnostic medical applications that employ EMFs (Hand et al., 2008).
The computational electromagnetics techniques have been enormously improved in the last years thanks to the progress in high-performance calculation, computer technology and development of acceleration software. The solution is based on commercial or custom-made codes based on various approaches such as the finite element method (FEM) or finite difference time domain method (FDTD), which allow to calculate the propagation of EMF waves (Wiart, 2016). In parallel, thanks to the improvements in medical imaging, computable humans’ models with high-number and high-resolution tissues can be used to conduct the exposure assessment to EMF in computational simulations. An example of possible high-resolution whole-body computational models is represented by the “Virtual Population” family by IT’IS foundation (Christ et al., 2009; Gosselin et al., 2014). It this way, it is currently possible to calculate case by case with high precision and accuracy the quantity of interest in specific tissues or organs of human models.
So, both experimental and computational methods are two ways to approach the same problem, but they are both reliable. Indeed, there is a huge literature of simulation works aimed to assess the exposure due to different exposure scenarios, without a reference to an experimental study. Our study uses a computational approach and is based on state-of-the-art human models (the Virtual Family) and a commercial and widely used simulation software (Sim4life. The word Sim4life searched in Scopus gives 157 documents in the period 2018-2022).
Point 5: An whole statistical analysis should be added. In particular, which one is the estimated error?
Response 5: We would thank the reviewer for the relevant comment. Following the suggestion of the reviewer we have performed a statistical analysis to compare the distribution found for the female and the male model for each of the wearable antenna positions. The statistical analysis confirms the difference in the trend of the distributions of SAR10g(p<0.05). A sentence on that additional analysis has been added in the revised paper:
Results, 224: “Moreover, a statistical analysis showed that the differences between the distributions of the SAR10g values of the male and the female models for each position of the wearable antenna resulted statistically significant (p-value < 0.01).”
As to the error, following the reviewer’s suggestion, we have included a paragraph in the “Discussion” session relative to the uncertainty:
- Discussion, 309: “However, it is noteworthy that in the interpretation of the results of EMF numerical dosimetry an intrinsic level of uncertainty, that can influence the reliability of the estimated SAR values, should be always taken into consideration. These uncertainties are difficult to be quantitatively estimated, but are qualitatively due to many factors, such as the reduced knowledge of dielectric properties of the human tissues, numerical limits of modeling of the source and anatomical body, discretization error, accuracy, and choice of the algorithm for SAR calculation [23-26]. In any case, although with unknown level of uncertainty, the numerical approach provides important and additional information on the distribution inside biological systems and organs that cannot be obtained only by experimental dosimetry [27].”
Point 6: An additional text about the SAR meaning and its side effect on human tissue should be added.
Response 6: Firstly, we would thank the reviewer for giving us appropriate and useful comments. We have noted the lack of the definition of the Specific Absorption Rate (SAR) in the original manuscript, so we have added a brief comment and the reference to the ICNIRP Guidelines in which the formula is reported.
- Materials and Methods, 143: “; it can be defined as the electromagnetic energy absorbed by a human tissue. With more details, the SAR is defined as the time derivative of the incremental energy consumption by heat involved in an incremental mass in a volume element, characterized with its density ρ and its conductivity σ [14].”

Reviewer 3 Report
- The novelties of the paper could be considered limited. The authors should try to more prominently present the new items of their work.
- Several additional results could be included in the numerical results section. For example different angles of exposure, or a variety of antennas. In fact, there are several other studies that perform such an investigation.
- Please provide some numerical evidence regarding the simulations via the FDTD method.
- It would be nice if the authors could compare their results by simulating their cases via the CST computational package.
Author Response
Point 1: The novelties of the paper could be considered limited. The authors should try to more prominently present the new items of their work.
Response 1: We thank the reviewer for the accurate revision work and the valuable feedback provided, giving us the opportunity to improve our paper.
The main aim of our paper was to further broaden the knowledge on the assessment of the exposure of the human being to the EMF emitted by a wearable antenna. In particular, our aim was to investigate the effects on the human exposure due to the position of the wearable antenna and to the interindividual human variability. There are different aspects of novelty, such as the antenna positions here studied and the use of anatomical model of both genders highly detailed. Indeed, we have analyzed 6 different antenna positions, whereas in most of the previous works the antenna was placed or near the head or on the trunk. We have also analyzed the effect of the helmet on the exposure level. Moreover, we considered for all these positions two different anatomical models, one male and one female, in order to assess the effect of the human variability also connected with the gender on the human exposure. Another aspect of novelties is the way in which the data are analyzed: instead of looking only at the peak level (which is an important aspect in the exposure assessment), we have also analyzed the distributions of the SAR10g in terms of descriptive statistics and percentage of point above a specific threshold, which is another innovative aspect.
In order to better present these items of our work, we have added some sentences in the Introduction (see revised paper and below).
- Introduction, 60: “The proposed study takes place in this context and our aim is to further broaden the knowledge on the assessment of the exposure of the human being to the EMF emitted by a wearable antenna. More in details, EMF human exposure assessment has been evaluated by means of electromagnetic computational method considering the wearable antenna posed in six different positions on the human body, mimicking different realistic exposure conditions. This will allow to evaluate the effect of the antenna position on the human exposure, in order to understand if and how the variation of the involved anatomical district could significantly influence the exposure. Moreover, to evaluate how the interindividual anatomical variability could impact on the level of exposure, two different human models were used, one of an adult male and one of an adult female. Indeed, there are differences in the anatomy of the models depending on the gender that could impact on the exposure level.”
Point 2: Several additional results could be included in the numerical results section. For example different angles of exposure, or a variety of antennas. In fact, there are several other studies that perform such an investigation.
Response 2: We thank the reviewer for the relevant comment and the useful suggestion.
The suggestion of the reviewer is interesting; however, it is out of the scope of our paper. Indeed, our aim was to emphasize the effects of the anatomical variability due to the differences between the human models and the effect of the variation of the position of the antenna among several anatomical districts. Nevertheless, we have preliminarily performed a set of simulations in which we have varied the coordinates of the antenna around the position described in the original manuscript and we have obtained no evident deviations from the reported results. In a following paper we will approach this other exposure variability aspects with the novel method of the stochastic dosimetry (i.e., Bonato et al., 2022; Chiaramello et al., 2021) in which the variability of the position and the angle of exposure of the antenna can be identified as input parameters of the surrogate model describing the exposure.
Point 3: Please provide some numerical evidence regarding the simulations via the FDTD method.
Response 3: Thanks to the reviewer for the appropriate comment. In order to provide a full answer to this comment, it is necessary to cite the Taflove et al. study (Taflove, A.; Hagness, S.C. Computational Electrodynamics: The Finite-Difference Time-Domain Method. 3rd ed. Norwood, MA: Artech House, 2005) in which the FDTD method is explained. This method was elaborated specifically for the electromagnetic problem, and it is the gold standard for the computational solution of the Maxwell’s equations. In order to better explain how the Finite-Difference Time-Domain works, we have added in the revision version of the manuscript the following sentence:
- Materials and Methods, 102: “All simulations were implemented with the finite-difference time-domain (FDTD) solver of the Maxwell’s equations through the approximation to finite differences. Briefly, the FDTD method involves both a spatial and temporal discretization of the electric and magnetic fields over a period of time and a specific spatial domain limited with the boundary conditions. Typically, the minimum spatially sampling is at intervals of 10-20 per wavelength, and temporal sampling is sufficiently small to maintain stability of the algorithm [16,17].”
Point 4: It would be nice if the authors could compare their results by simulating their cases via the CST computational package.
Response 4: First of all, we would thank the author to have given us the opportunity of deepening the question about the comparison between several different software. Indeed, in the “Discussion” section of the original manuscript we have not specified the software that have been used in the cited studies. In our session, we have reported, for example, the study performed by Ali et al. [11] in which the CST Microwave Studio software has been used for the simulation of the phantom with the wearable antenna posed on the top; their results agree with our ones, although the simulated scenario is not exactly the same and the simulation software are different.
The Sim4life software used in our study is a well-known commercial software used in the field of exposure assessment and it is a specific support for the study of the interactions between the EMFs and the human being. It has been already validated and used in many exposures assessment study and biomedical dosimetric study (i.e., a search on Scopus with the word Sim4life gave 157 documents in the period 2018-2022), therefore there is no need to perform the same simulations with another computational software to validate it. Moreover, CST is not a golden standard and, unfortunately, is a software for which a license is needed.

Round 2
Reviewer 1 Report
> After reviewing the comments from the authors and the modifications> to the paper, I feel that the scope of the paper is too narrow to> give it much value. It seems that the focus of the article is an> exercise on using software for simulations, while avoiding the real> problem of human exposures to radiation. Publications on how> exposures from antennas on the skin can be reduced would be much> more valuable.>> Reject.
Reviewer 3 Report
The authors have improved their work.